# Utilization of social media communities for caregiver information support in stroke recovery: An analysis of content and interactions

Elton H. Lobo[1,2]*, Tara Johnson[2,3], Anne Frølich[2,4], Finn Kensing[5], Lene J. Rasmussen[6,7], Sarah M. Hosking[3], Amy T. Page[8,9], Patricia M. Livingston[3], Sheikh Mohammed Shariful Islam[10], John Grundy[11], Mohamed Abdelrazek[1]

1 School of Information Technology, Deakin University, Geelong, VIC, Australia, 2 Department of Public Health, University of Copenhagen, Copenhagen, Denmark, 3 Faculty of Health, Deakin University, Geelong, VIC, Australia, 4 Innovation and Research Centre for Multimorbidity, Slagelse Hospital, Slagelse, Region Zealand, Denmark, 5 Department of Computer Science, University of Copenhagen, Copenhagen, Denmark, 6 Department of Cellular and Molecular Medicine, University of Copenhagen, Copenhagen, Denmark, 7 Center for Healthy Aging, University of Copenhagen, Copenhagen, Denmark, 8 Pharmacy Department, Alfred Health, Melbourne, VIC, Australia, 9 Centre for Medicine Use and Safety, Monash University, Melbourne, VIC, Australia, 10 Institute for Physical Activity and Nutrition (IPAN), Deakin University, Geelong, Australia, 11 Faculty of Information Technology, Monash University, Melbourne, VIC, Australia

* elobo@deakin.edu.au

**Data Availability Statement:** All relevant data are within the paper and its Supporting Information files.

## Abstract

### Background

Caregivers often use the internet to access information related to stroke care to improve preparedness, thereby reducing uncertainty and enhancing the quality of care.

### Method

Social media communities used by caregivers of people affected by stroke were identified using popular keywords searched for using Google. Communities were filtered based on their ability to provide support to caregivers. Data from the included communities were extracted and analysed to determine the content and level of interaction.

### Results

There was a significant rise in the use of social media by caregivers of people affected by stroke. The most popular social media communities were charitable and governmental organizations with the highest user interaction–this was for topics related to stroke prevention, signs and symptoms, and caregiver self-care delivered through video-based resources.

### Conclusion

Findings show the ability of social media to support stroke caregiver needs and practices that should be considered to increase their interaction and support.

**Funding:** This study was supported through doctoral scholarships from the School of Information Technology, Deakin University, and the Department of Public Health, University of Copenhagen. Further, Prof John Grundy is supported by ARC Laureate Fellowship FL190100035, and Dr. Sarah Hosking is supported by an Alfred Deakin Postdoctoral Research Fellowship.

**Competing interests:** The authors have declared that no competing interests exist.

**Abbreviations:** QoL, Quality of Life; ADL, Activities of Daily Living; HCP, Health Care Professionals.

# Introduction

Stroke is the leading cause of dependency and disability worldwide [1], resulting in family caregivers providing substantial care to people with stroke [2]. Family caregivers, generally known as informal caregivers [3] are responsible for assisting with daily activities, including mobilization, toileting, bathing, transportation, and navigating the health care system [4]. Despite their key role in care, many caregivers feel unprepared [5], leading to psychological, social, physical, and financial strains [6].

Family (or informal) caregivers have varying needs for education and support during the stroke care trajectory [7]. Yet, to date, standard clinical practice guidelines have not considered programs to ensure caregiver education and support [8]. The most common form of information received by caregivers at the hospital included booklets and pamphlets, which caregivers have reported to be very basic or out of date [9]. Caregivers may attempt to source alternate information sources to improve preparedness to reduce uncertainty and enhance recovery [10].

In the past, caregivers have predominately used the internet (or online) sources to access information related to stroke care [11–13]. The internet is changing how health information is accessed [14], thereby influencing individuals' knowledge, attitudes, and beliefs towards a specific health behaviour [15]. As a result, the trend towards internet use for health information purposes has been significantly rising [16]. A cross-sectional study by Naqvi, Montiel [17] reported over 96.8% of caregivers having access to the internet to generally browse web pages (84.6%) and access their emails (89.4%).

Today, in the era of Web 2.0, social media such as Facebook and Twitter has changed the landscape in health care information delivery [16, 18]. Social media can empower people to adopt a healthy lifestyle and help improve health management and decision-making processes [19]. Furthermore, social media creates an unprecedented opportunity to enhance the quality of care by mobilizing many social media users and enabling the users to generate a large amount of content [19]. The content generated is in the form of user health care knowledge, experiences, symptoms, health care products, doctors, and medicines in easily accessible formats, such as images, text, and videos [18].

Social media use has provided organizations and individuals with an openly accessible platform to engage actively and participate in healthcare [20]. However, very little is known about its potential benefit to caregiving and its ability to interact with the caregiver actively. This study presents three key aims. This study aims to:

1. Investigate frequency of searches for stroke-related terms over time using Google Insights and Google Trends.

2. Identify the information content available to caregivers on popular information-support-based social media platforms (i.e., Facebook and Twitter) to support their needs and activities.

3. Understand the levels of interaction for the different social media posts identified through the likes, comments, and shares by content types (i.e., image, video, link, or text).

# Method

## Study design

Our study consisted of a mixed-method approach to answer identified research aims. The mixed methods approach is a type of research where a researcher or group of researchers combine elements of quantitative and qualitative methods (e.g. use of quantitative and qualitative

viewpoints for data collection, analysis and inference techniques) to provide a broad understanding of the research problem [21]. For example, to investigate the frequency of stroke terms, a quantitative analysis was conducted to determine the online activity of people interested in stroke recovery and care using tools such as Google Trends and Google Insights. Google Trends and Google Insights provide a platform for individuals to investigate its users' search behaviour throughout time based on a relative cumulative search volume score from 0–100, which is the ratio of single search term volume to all possible searches. A qualitative analysis was used to analyse the information content using a thematic synthesis approach. Finally, the levels of interaction were identified through a quantitative statistical analysis of likes, comments, and shares based on the different content types.

## Identifying relevant communities

The identification of relevant social media communities (or groups) in stroke recovery involved multiple steps. Initially, we identified the relevant search keywords used based on discussions with topic experts and electronic database searches. We tested the keywords on Google Insights and Google Trends to determine their relevance to individuals around the world in stroke recovery and care based on their searching behaviour. Finally, we performed individual searches on two popular social media platforms (i.e., Facebook and Twitter).

A search of social media platforms (i.e., Facebook and Twitter) was conducted from December 2020 to January 2021 and was limited to those available in the English language. Moreover, the search included only communities made public by the administrator (or did not require permissions to be accessed by the user).

## Community selection

Initially, the researchers used a custom-built web form to manually extract information from all social media communities, including community names, descriptions, links, number of followers (or likes), and several posts, and store the data in a MySQL database. The communities identified were then filtered based on the inclusion and exclusion criteria described in **Table 1**.

## Analysis the findings

The analysis process involved a multi-stage data extraction and management process using a custom-built python scraper consisting of all the community page links and outputs findings to a MySQL database. The data was then extracted as a Microsoft Excel file and coded independently using QSR NVivo 12 by two researchers based on a three-stage thematic synthesis approach, involving: 'line by line' coding of text, development of descriptive themes, and generation of analytical themes [22]. All posts unrelated to the caregiver and/or posts that did not provide information support (e.g., advertisements, event photos, news articles, research studies, etc.) were excluded from the study. Additionally, descriptive characteristics data from the communities (such as community name, origin, published date, and basic information) and interaction data (such as likes and comments) were charted by one researcher to answer the specific research aims.

**Table 1. Inclusion and exclusion criteria used to filter social media communities.**

| Inclusion Criteria | Exclusion Criteria |
| --- | --- |
| • Considers Caregivers of Stroke described through its description or content<br>• Provides Information regarding Stroke<br>• Supports User Interaction on Posts | • Does not include Caregivers of Stroke<br>• Blocks Users from Replying to Posts |

## Collating and summarizing

Both qualitative and quantitative findings were collated and summarized to answer the research questions resulting in the descriptive numerical summary and thematic analysis. The predefined descriptive classification applied to the initial coding of the communities include;

a. **Community Demographics**

- *Year Published*–to understand the growth in online communities over the past few decades.

- *Community location*–to understand the target population

- *Community affiliation*–to know if the content created is by people working in the stroke domain

b. **Community Purpose**–to understand the purpose of the community through the community description

c. **Information Support**–to understand the type of information provided to the caregiver in the post (i.e., disease, patient care management, self-care, etc.) and the method of delivery (i.e., text, image, video, or link) using a thematic analysis technique

d. **Community Interaction**

- *Post purpose*–to understand the information type required by the user

- *Likes, followers, reactions, and comments*–to understand user interaction based on the post purpose

## Results

### Digital interest regarding stroke

Overall, 94 keywords were identified from discussions with topic experts and electronic database searches. Of these 94 keywords, 15 keywords were based on stroke disease and its definitions, 25 keywords were related to the signs & symptoms of stroke, 37 keywords included different medications used in stroke and 17 keywords focused on aspects related to recovery & care.

Findings from the Google Trends and Google Insights searches demonstrated an apparent increase in the cumulative search volumes for the terms identified through discussions with topic experts and electronic database searches over the past ten years (**Fig 1**). The rise in the cumulative search volume was 12.4 between January 2011 and December 2020 identified by:

$$a_{ij} = \frac{\sum k_i}{N_{ij}} \tag{1}$$

Where $a_{ij}$ is the average cumulative search volume for each topic ($j$) each year ($i$), $k$ is the cumulative search volume acquired from Google Trends and Insights for all the keywords associated with the topic for year $i$, $N$ is the total number of keywords in the topic ($j$) for year $i$, $i$ is the year ranging from 1 to 10 and $j$ is the topic ranging from 1 to 4.

$$Y_i = \frac{\sum_{j=1}^{j=4} a_{ij}}{4} \tag{2}$$

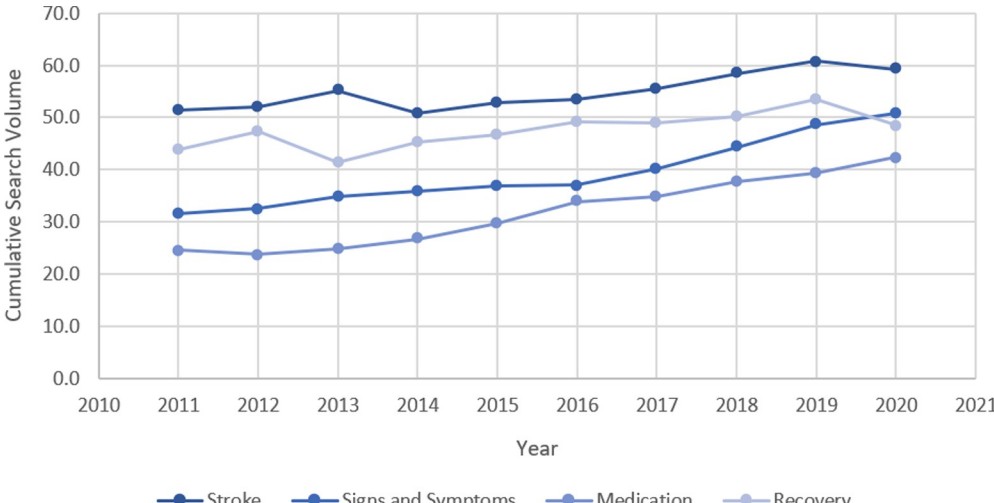

**Fig 1. Cumulative search volume from January 2011 to December 2020 for stroke related topics used in this study.**

where $Y_i$ is the average cumulative search volume for all topics ($a_{ij}$) in year $i$ ranging from 1 to 10

$$A = Y_{10} - Y_1 \tag{3}$$

where $A$ is the rise in the cumulative search volume between the Tenth ($Y_{10}$) and First ($Y_1$) Years (i.e. January 2011 and December 2020).

Topics including 'stroke definition' and 'stroke recovery' were the most commonly searched during the ten years. Issues such as 'signs & symptoms of stroke', and stroke medication have had a significant rise in searches during the past four years.

**Identification of relevant keywords.** Table 2 presents the ten most commonly searched terms identified by the online search (or usage) activity as extracted from Google Trends and Google Insights. The ten most frequently used keywords were selected based on their cumulative search volume over the past year.

**Table 2. Ten most commonly used keywords in stroke identified through Google Trends.**

|  | Cumulative Search Volume |
|---|---|
| Stroke | 85.3 |
| Stroke Care | 81.6 |
| Stroke Recovery | 76.8 |
| Apoplexy | 76.3 |
| Cerebrovascular Accident | 75.6 |
| Stroke Unit | 75.1 |
| Traumatic Brain Injury | 74.5 |
| Lacunar Infarct | 74.2 |
| Stroke Medication | 73.6 |
| Aphasia | 73.1 |

## Social media communities

The combined search strategies identified 352 social media communities using the keywords identified in **Table 2**, which were then screened for eligibility using the inclusion and exclusion criteria demonstrated in **Table 1**. Out of the 352 social media communities, 111 were excluded as they were not related to stroke patient caregivers, 23 were not accessible to the public, 17 were not associated with stroke, and 3 were not available in English. Overall, 198 social media communities were included in the study, as shown in **Table 3**.

**Descriptive characteristics.** Out of the 198 social media communities, 141 (71.2%) were available on Facebook and 57 (28.8%) were available on Twitter. These communities were created by individuals (n = 64; 32.3%), charitable or non-profit organizations (n = 61; 30.8%), community centres (n = 24; 12.1%), educational organizations (n = 20; 10.1%), medical centres (n = 13; 9.1%), small and medium sized organizations (n = 8; 4.0%) and governmental organizations (n = 3; 1.5%) identified based on administrator affiliations and community descriptions as illustrated in **Fig 2**. The most popular groups, identified by the number of followers, were charitable organizations and governmental organizations (**Fig 3**).

Across all social media platforms, Twitter was seen to have the highest average number of followers and posts (7093.6 followers and 4828.7 posts), followed by Facebook (4202.8 followers and 579.6 posts) as shown in **Fig 4**. The earliest identified pages were published in 2009 on both Twitter (n = 10; 5.1%) and Facebook (n = 5; 2.5%). Since 2009, both social media platforms have witnessed a variation in the number of new stroke communities for caregivers (**Fig 5**).

**Community purpose.** The analysis of the social media community description identified six prominent themes (**Fig 6**) detailed below:

a. **Support:** Support-based social media communities were the most common community type (n = 81; 40.9%); these are intended to provide users with tools to support and share caregiving responsibilities. Moreover, these communities allowed users to join either virtual or local groups to promote emotional and psychological support.

b. **Awareness:** Communities in this theme (n = 59; 29.8%) intend to make the caregiver more aware of the tools and resources available locally to support the patient during care. It also allowed the caregiver to understand the risk factors and signs of a stroke to prepare them during a secondary stroke event.

c. **Education:** The education theme (n = 34; 17.2%) consisted of communities that share online books and resources intended to educate the caregiver on stroke-related topics, factors associated with its occurrence, secondary prevention techniques, management, support guidelines, medication resources, and similar issues. This was generally delivered in the form of text and video-based resources.

d. **Advertising:** These communities (n = 14; 7.1%) generally focused on advertising recovery products to support caregivers during care and ongoing research conducted at local universities to develop better care practices to support caregivers and their patients.

e. **Motivation:** Motivation (n = 8; 4.0%) oriented communities generally delivered this by caregivers and patients through personal stories and practices during recovery. Social media communities motivated their users through inspirational quotes and success stories.

f. **Fundraising:** The fundraising communities (n = 2; 1.0%) were either delivered by charitable organizations to support caregivers and their patients or by individual caregivers struggling to support patients due to financial constraints. The fundraising in charitable organizations involved links to fundraising campaigns and campaign invites to events conducted locally.

**Table 3. Social media communities included in the review.**

| Facebook (n = 169) | |
|---|---|
| • Association for the Rehabilitation of the Brain Injured | • Minnesota Brain Injury Alliance/Minnesota Stroke Association |
| • Stroke Recovery Foundation | • Suncoast Aphasia Support Group |
| • United Stroke Alliance | • Oceanside Stroke Recovery Society |
| • American Stroke Association | • Orillia Stroke Survivor and Caregiver Support Group |
| • Stroke Association | • Delta Stroke Recovery Society• Pittsburgh Aphasia Community |
| • Stroke Foundation | • StrokeEd |
| • National Aphasia Association | • Aphasia Lab-USC |
| • Aphasia Recovery Connection | • BRAIN Lab: Brain Research for Aphasia and Intensive Neurorehabilitation Lab |
| • American Stroke Foundation | • Aphasia CRE |
| • Stroke Association NI | • STROKE-The Road to Recovery |
| • Stroke Association South West | • World Stroke Campaign |
| • Stroke Survivors Foundation | • Stroke Special Interest Group |
| • Aphasia Ireland | • University of Michigan Aphasia Program (UMAP) |
| • Stroke Association East of England | • Stroke Rehabilitation Research |
| • Aphasia Network | • Stroke and Cerebrovascular Accident Education |
| • Aphasia Nova Scotia | • Triangle Aphasia Project, Unlimited |
| • Stroke Association London | • Aphasia Connections |
| • Stroke Family Awareness | • Priority Research Centre for Stroke and Brain Injury |
| • Bright Spot Pediatric Stroke | • The Big Sky Aphasia Program |
| • American Aphasia Society | • Purdue University Aphasia Group |
| • FAST Stroke Awareness | • Hazard & Surrounding Area Stroke Survivor & Caregiver Support Group |
| • World Stroke Day Kenya 2017 | • Spot Stroke |
| • Stroke SA Inc | • Kathi Naumann -Stroke Support & Survival Guide |
| • Supporting Aphasia Fellowship and Education Fellowship and Education | • The Aphasia Cafe by Dr. Dawn McGuire |
| • Brain Injury Recovery Foundation | • Stroke Awareness |
| • Australian Aphasia Association | • Raising Stroke Awareness |
| • Stroke Foundation of NZ | • Stroke Awareness for Everyone |
| • Think Ahead Stroke | • Stroke Prevention |
| • Stroke Fighters | • Aphasia Awareness |
| • Stroke Survivors Empowering Each Other (SSEEO) | • Stroke therapy tricks for stroke survivors |
| • Singapore National Stroke Association | • Stroke Group |
| • BINA Stroke & Brain Injury Assistance | • Canadian Aphasia Association |
| • Stroke Rehabilitation & Healing, Inc. | • Aphasia Awareness |
| • Calgary Aphasia Centre | • Stroke |
| • Stroke Support of Texas | • Stroke Cure |
| • Stroke Help Network | • Stroke Rehabilitation Awareness |
| • Aphasia NSW | • Stroke Caregiver |
| • The Scott Coopersmith Stroke Awareness Foundation | • Rehabilitation for Stroke |
| • Brain injury & Stroke Foundation KENYA | • TBI Hope & Inspiration |
| • Friends of Aphasia | • The Brain Fairy—Living with Brain Injury |
| • Retreat & Refresh Stroke Camp | • Aphasia Friendly Resources |
| • Adler Aphasia Center | • Stronger After Stroke Blog |
| • Aphasia Center of California | • Stroke information |
| • Living with Aphasia | • Stroke Support |
| • Talkback Association for Aphasia Inc | • Recovering from Brain Injury |
| • Stroke Information Support Group | • Stroke Recovery Tips |
| • Alberta Aphasia Camp | • Stroke Recovery: Stories from Patients, Doctors, Families and Caregivers |
| • Aphasia Centre of Ottawa | • Stroke |
| • Aphasia vzw | • GRASP—Geriatric Relearning After Stroke-Induced Paralysis |
| • Stroke Rehabilitation Ireland | • Caregiving for Stroke Survivors |
| • Stroke Caregivers | • Teamconnor fundraising and brain injury/stroke awarness |
| • Stroke Ownership & Recovery | • Stroke Survivor Caregivers |
| • Midwest Stroke support group for survivors and caregivers | • Surviving A Stroke |
| • Stroke,tbi,and their,caregivers | • Stroke Survivors |
| • The Other Stroke Talk for survivors, caregivers and anyone who wants to be | • Caregiving After Stroke |
| • Support for Caregivers of Stroke Patients | • Stroke Awareness |
| • Malaysian Stroke Rehabilitation | • Stroke Recovery KW |
| • Stroke & Neuro Intervention | • Stroke Warriors |
| • Aphasia SG | • Stroke Survivor |
| • UCAN Stroke Rehabilitation in Merseyside and Cheshire | • Stroke Rehabilitation |
| • Stroke Support India | • TBI & Stroke Victims |
| • AphasiaAccess | • Stroke Recovery Solutions |

*(Continued)*

**Table 3.** (Continued)

| | |
|---|---|
| • Stroke Survivors<br>• Certified Stroke Rehabilitation Specialist (CSRS)<br>• Greenhills Stroke Rehabilitation Center Ghana<br>• Stroke Rehabilitation<br>• Montgomery County Stroke Survivor, Caregiver, and Aphasia Support Group<br>• Stroke & Neuro Rehabilitation for Shropshire<br>• Stroke Rehabilitation Centre<br>• UNT Aphasia Support Group<br>• Stroke Awareness | • Aphasia will not be silent / Stroke Survivor Coach<br>• Stroke UK<br>• TBI TED—Brain Injury and CTE Support<br>• Group Stroke<br>• Stroke Therapy<br>• Stroke Rehab<br>• NXT Senior & Caregiver Resources Inc. |
| **Twitter (n = 29)** | |
| • American_Stroke<br>• Stroke Association<br>• Sign Against Stroke<br>• heartandstroke<br>• Aphasia Hope<br>• Stroke Foundation<br>• Croi- Heart & Stroke<br>• American Heart News<br>• Better Conversations<br>• davida godett<br>• Million Hearts<br>• Tactus Therapy<br>• ARC AphasiaRecovery<br>• HeartFoundationSA<br>• Northern Ireland Chest Heart & Stroke<br>• Stroke Association Yorkshire<br>• BAS<br>• INS<br>• Prasanna Tadi M.D TEDx Speaker, Stroke Doc, Blogge<br>• Natl Aphasia Assoc<br>• Heart&Stroke NB \| Coeur+AVC NB<br>• Aphasia Institute<br>• LivingWithAphasia<br>• Heart & Stroke Science<br>• Treat The Stroke<br>• Aphasia Nova Scotia<br>• Stroke Connection<br>• Connect<br>• BIAAZ | • CDC Division for Heart Disease & Stroke Prevention<br>• The Aphasia Center<br>• Stroke Survivors Foundation<br>• Adler Aphasia Center<br>• Stroke Foundation NZ<br>• Caregiver's Cargiver<br>• StrokeRehab Plymouth<br>• Stroke Recovery<br>• Dyscover<br>• East Lancs Stroke Assistance & Support<br>• Heart & Stroke NL<br>• fermanagh Stroke Support Group—SOSS<br>• Reclaiming Ourselves<br>• StrokeSupport<br>• Stroke Recovery Association MB<br>• Stroke Support Group<br>• act F.A.S.T<br>• Stroke Rehab<br>• City Access—Resources for Aphasia<br>• Stroke Recovery Association NSW<br>• StrokeSmart Magazine<br>• IschemicStroke<br>• BIA-MA<br>• Stroke Caregivers<br>• Signs Of Stroke<br>• Stroke Support<br>• BIAF<br>• BrainLine.org |

**Community role in information support.** While community information extracted demonstrated a total of 356,960 posts, only 173,508 posts could be extracted using a python-based scraper tool. Of these 173,508 posts, the following posts were excluded: 6369 (related to motivating the individual), 16960 (focused on advertising local events, products, and research), 45726 (consisted of news articles regarding stroke), 25939 (included photos or videos of local community activities or events), 28089 (focused on creating awareness for the prevention of the disease), 4176 (looked to fundraise to support an individual or organization), 24672 (did not provide information support), and 14070 (did not offer general stroke information or focus on caregivers). The remaining 7507 posts provided the caregiver with information to support them during the care trajectory, and hence were further analysed and classified as summarized in **Table 4**.

**Analysis of interaction.** **Table 5** summarizes user interaction based on the topics identified in **Table 4** and content type (i.e., text, image, video, and link), identified through the average of likes, shares, and comments. The data presented showed that the individual's interaction with the post varied based on the topic and the content type. For example, the

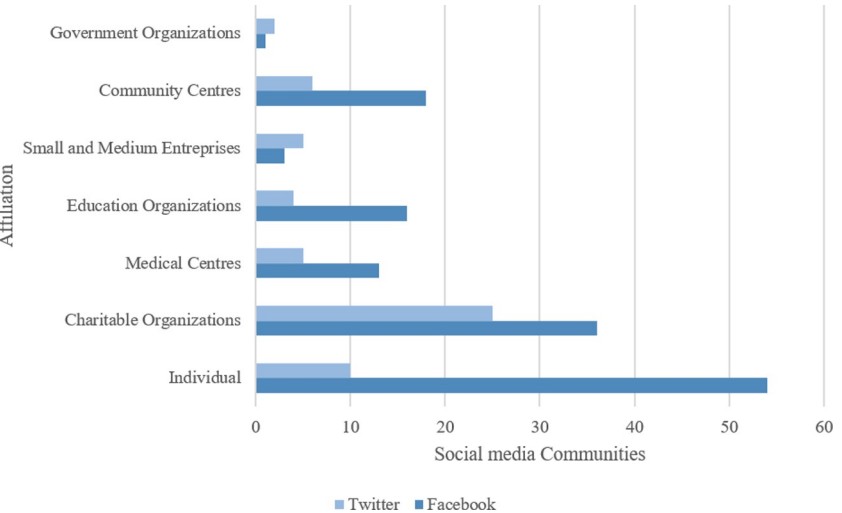

**Fig 2. Social media communities by affiliation.**

target user group generally interacted with video-based content (i.e., Likes– 13.41, Comments– 8.79 and Shares– 8.53) followed by image (i.e., Likes– 12.35, Comments– 4.46 and Shares– 6.69), link (i.e., Likes– 6.99, Comments– 1.59 and Shares– 3.06) and text (i.e., Likes– 4.03, Comments– 1.68 and Shares– 2.58) based content as shown in **Fig 7**. While the most interacted topics based on content type has been illustrated in **Fig 8** identified through the data summarized on **Table 5**.

## Discussion

This study aims to highlight the information-seeking behaviour of people affected by stroke and the interaction of content created for caregivers on popular social media platforms (i.e., Facebook and Twitter). This study is significant for content creators of social media communities to identify appropriate topics to support stroke caregiving needs and promote caregiver

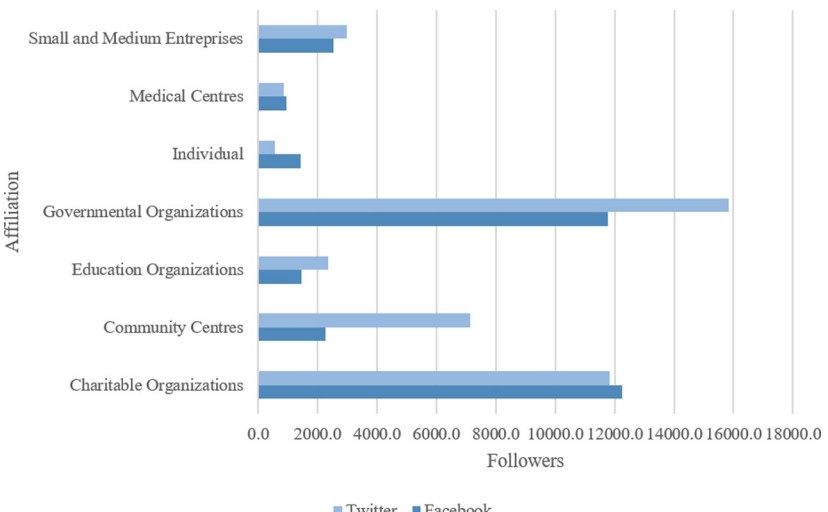

**Fig 3. Social media communities by followers and affiliation.**

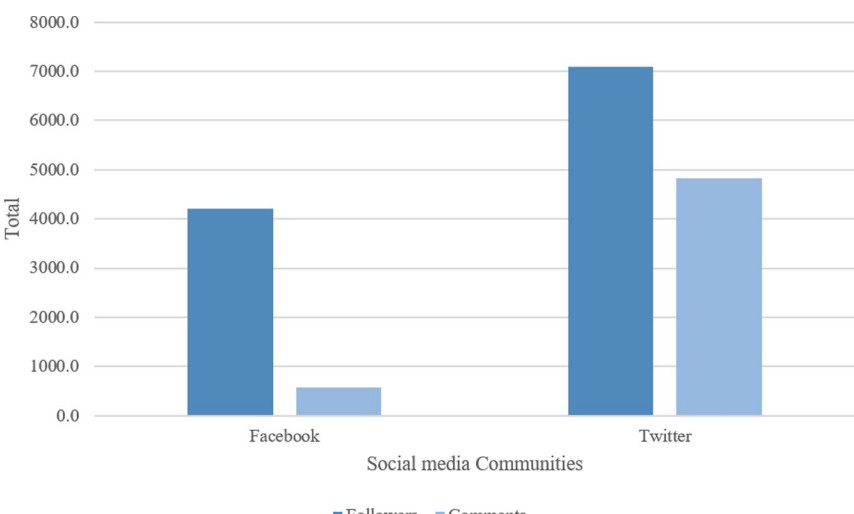

**Fig 4. Social media communities by followers and posts.**

interaction within the community, thereby ensuring caregiver education and preparedness when supporting the survivor.

Findings from our Google Insights show an increase in search trends for stroke-related topics over the past ten years. The growth has been predominately for topics related to the signs and symptoms and medications, with stroke definition and recovery being the most popular searches over the past ten years. This concurs with Tan and Goonawardene [23], which suggests an increase in users seeking health information online to ensure education and preparedness for the disease, thereby allowing them to make better healthcare decisions during recovery.

The increase in user access to internet resources for stroke was not limited to Google searches but also within popular social media platforms. The findings from the study show an increase in social media communities for caregivers post-2009 created by individuals with different affiliations. A majority of which are individuals and charitable organizations. However,

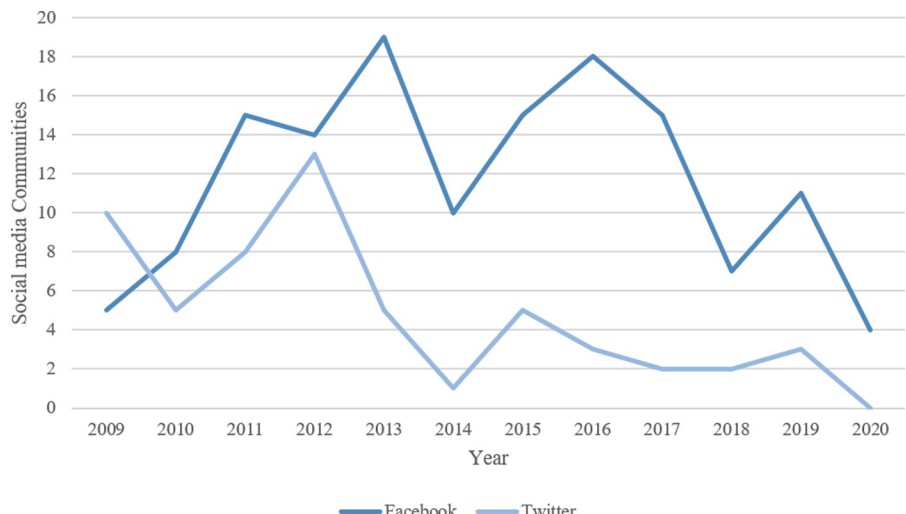

**Fig 5. Variations in new social media communities by year.**

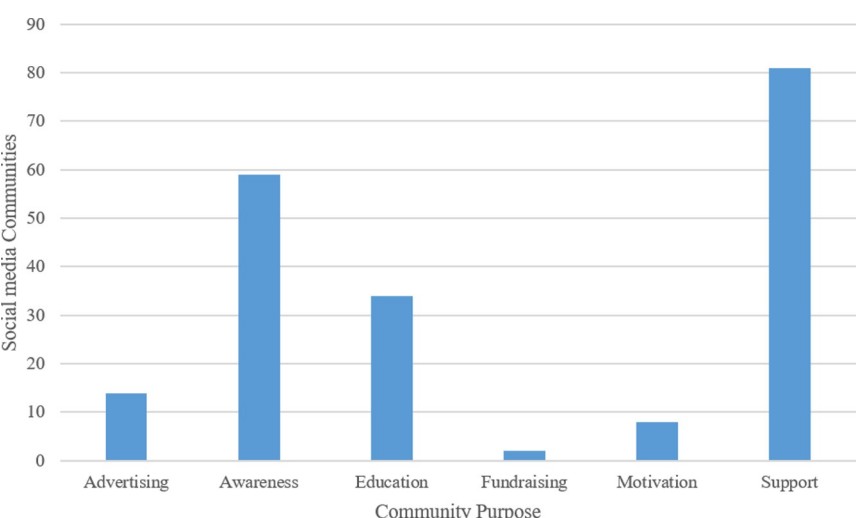

**Fig 6. Social media communities by community purpose and followers.**

the most accessed social media communities were found to be affiliated with governmental and charitable organizations. This could be due to the trust factor associated with information provided by federal agencies and community organizations, as highlighted in the study by Dutta-Bergman [24], suggesting that the information provided by these individuals is based on expert-based literature and credible sources.

Nowadays, misinformation or lack of quality information is a growing problem [25]. Crocco, Villasis-Keever [26] in a systematic review highlighted the internet's capacity to harm the health of the user to be equal to the good and useful information it provides in a relatively timely and inexpensive manner. For example, in one case the misinformation available on the internet contributed to emotional harm, while in another case lead to hepatorenal failure in an oncology patient who obtained misinformation regarding medication use over the internet [26]. To prevent healthcare issues and fears amongst the population, Cuan-Baltazar, Muñoz-Perez [25] suggests the need for governmental organizations to develop a strategy that teaches its residents to verify the quality of information they read. Moreover, Swire-Thompson and Lazer [27] describes the need for internet users to collaborate with physicians to ensure they are more actively involved in the decision-making processes, and they are aware of methods to separate health myths from facts that the internet provides.

While social media communities do not exclusively focus on the caregiver, it was possible to identify the relevant communities and posts through their content, which were classified in this study based on their relevance. The classification involved two categories; (i) General, i.e., posts that enabled the caregiver to understand the disease, causes, types, diagnosis methods, risk factors, prevention, consequences, and treatment, and (ii) Caregiver, i.e., information to enable the caregiver to communicate with relevant stakeholders, understand the impact of caregiving, understand the roles and decision making practices, understand means to support and care for the patient and to ensure self-care. Overall, findings from these comments high-light a positive interaction in terms of likes, shares, and comments, especially for video-based content and topics related to prevention, self-care, signs and symptoms, caregiver impact, and patient support and care.

Video-based education resources have numerous advantages to promote positive health decisions and lifestyle changes [28]. The benefits include: (i) cost-effectiveness, (ii) removal of

**Table 4. Topics identified and their frequency of occurrence on the two social media platforms.**

| Information Topics | Frequency | Percentage (%) |
|---|---|---|
| GENERAL | 4609 | 66.90 |
| What is Stroke? | 89 | 1.29 |
| Signs and Symptoms | 753 | 10.93 |
| Types of Stroke | 112 | 1.63 |
| Causes of Stroke | 54 | 0.78 |
| Diagnosis | 20 | 0.29 |
| Risk Factors | 1011 | 14.68 |
| Demographics | 122 | 1.77 |
| Heart and Vascular Health | 714 | 10.36 |
| Mental Health | 46 | 0.67 |
| Women's Health | 30 | 0.44 |
| Lifestyle | 312 | 4.53 |
| Medication | 66 | 0.96 |
| Other Medical Conditions | 181 | 2.63 |
| Diabetes | 175 | 2.54 |
| Head Injury | 6 | 0.09 |
| Prevention | 1310 | 19.02 |
| Managing Lifestyle | 1236 | 17.94 |
| Managing Mental Health | 100 | 1.45 |
| Managing Medical Risks | 35 | 0.51 |
| Managing Sleep | 30 | 0.44 |
| Consequences | 758 | 11.00 |
| Cognitive | 613 | 8.90 |
| Emotional | 103 | 1.50 |
| Physical | 103 | 1.50 |
| Sleep | 8 | 0.12 |
| Relationships | 2 | 0.03 |
| Quality of Life | 2 | 0.03 |
| Living and Independence | 5 | 0.07 |
| Treatment | 1234 | 17.91 |
| Treatment Practices | 103 | 1.50 |
| Importance of Early Treatment | 83 | 1.20 |
| Rehabilitation | 820 | 11.90 |
| Guidelines | 454 | 6.59 |
| Importance | 28 | 0.41 |
| Cost | 4 | 0.06 |
| At-Home Rehabilitation | 363 | 5.27 |
| Treatment of Risk Factors | 305 | 4.43 |
| Monitoring | 136 | 1.97 |
| Surgery | 13 | 0.19 |
| Medications | 204 | 2.96 |
| CAREGIVER | 2280 | 33.10 |
| Impact | 184 | 2.67 |
| Communication Practices | 117 | 1.70 |
| Health Professional | 12 | 0.17 |
| Patient | 105 | 1.52 |
| Roles and Decision Making | 21 | 0.30 |

(*Continued*)

**Table 4.** (Continued)

| Information Topics | Frequency | Percentage (%) |
|---|---|---|
| Patient Support & Care | 1195 | 17.35 |
| Care Guidelines | 1077 | 15.63 |
| Supporting Activities of Daily Living | 123 | 1.79 |
| Finance & Legal Support | 72 | 1.05 |
| Care Planning | 162 | 2.35 |
| Self-care | 864 | 12.54 |
| Need | 81 | 1.18 |
| Strategies | 864 | 12.54 |
| Take a Break | 52 | 0.75 |
| Engage in Other Activities | 105 | 1.52 |
| Manage Quality-of-Life | 131 | 1.90 |
| Manage Health & Well-being | 677 | 9.83 |
| Manage Emotions | 46 | 0.67 |
| Manage Relationships | 48 | 0.70 |
| Sharing Care Responsibilities | 55 | 0.80 |

inconsistencies and presentation of information in a standardized format, (iii) creation of content that allows individuals with low health literacy to comprehend health information, and (iv) access through numerous different platforms or interventions [29]. However, Ferguson [28] highlights the importance of presenting the content concisely to avoid overwhelming the target audience with information, with a specific focus on the video length to ensure attentiveness of the target audience during the duration of the video.

While this study suggests caregivers in the stroke generally prefer video-based resources on social media communities, it is crucial to understand the influences of other media like text and images on health education. For instance, text-based resources allow individuals to access materials at their own pace and may be easier to access than video-based resources, particularly for individuals with low technical literacy [29]. On the other hand, images benefit individuals with low literacy skills [30] and have enhanced comprehension, satisfaction, and readability amongst the target audience [31].

Given that information type (i.e., video, image, and text) is a critical aspect for delivering information to specific individuals, it is also equally essential for one to consider individuals' needs to maximize interaction. Despite the existing set of topics that researchers believe to be important to address specific health information needs, there are several differences in the actual individual's needs [32]. For example, researchers are influenced by the disease type and researcher's motivation [32], while in stroke caregiving, the caregiver's needs differ based on the different stages of the survivors' illness, the need to maintain care continuum, and to ensure self-care during recovery [7, 33]. The need to maintain a care continuum and ensure self-care was evident in this study, with maximum interaction identified in prevention, signs & symptoms, patient support & care, risk factors, caregiver impact, and self-care. However, greater emphasis would need to be considered to provide information at different stages of the survivors' illness, which is currently not evident. In addition, it is important to understand the literacy and communication barriers that may impact the target audience and may limit their motivation to engage with the information, which can be restricted by co-designing information to limit these barriers [30].

One method that can be implemented when designing health information is Participatory Design (or PD) approach [34]. The PD approach has been drawn from several methods,

**Table 5. Analysis of user interaction based on the averages of likes, shares and comments for different content types.**

| Content | Type | Likes | Shares | Comments |
|---|---|---|---|---|
| *General* | Text | 4.12 | 1.83 | 2.07 |
| | Image | 13.36 | 4.35 | 6.5 |
| | Video | 14.73 | 9.44 | 9.03 |
| | Link | 7.39 | 1.73 | 3.49 |
| What is Stroke? | Text | 0.33 | 1 | 0.33 |
| | Image | 12.62 | 3.73 | 10.87 |
| | Video | 1 | 0 | 0.67 |
| | Link | 9 | 0.41 | 2.54 |
| Signs and Symptoms | Text | 17.87 | 3.1 | 2.57 |
| | Image | 13.29 | 3.6 | 11.61 |
| | Video | 19.42 | 7 | 13.02 |
| | Link | 9.77 | 3.96 | 8.11 |
| Types of Stroke | Text | 7.75 | 1 | 6.5 |
| | Image | 8.04 | 0.96 | 4.81 |
| | Video | 2.71 | 0 | 1.71 |
| | Link | 10.57 | 1.25 | 4.35 |
| Causes of Stroke | Text | 0.67 | 0.67 | 0.67 |
| | Image | 6 | 3 | 3.8 |
| | Video | 8 | 0 | 7.67 |
| | Link | 14.32 | 1.84 | 7.38 |
| Diagnosis | Text | 2 | 0 | 0 |
| | Image | 6 | 0 | 0.5 |
| | Video | 0.5 | 0 | 0 |
| | Link | 11.67 | 0.67 | 5.67 |
| Risk Factors | Text | 7.6 | 2.82 | 3.2 |
| | Image | 25.61 | 5.28 | 6.66 |
| | Video | 6.49 | 7.18 | 6.28 |
| | Link | 4.54 | 2.25 | 3.67 |
| Prevention | Text | 8.45 | 3.21 | 3.69 |
| | Image | 18.21 | 4.13 | 5.92 |
| | Video | 21.51 | 8.07 | 12.1 |
| | Link | 3.54 | 1.86 | 2.56 |
| Consequences | Text | 1.53 | 1.69 | 1.59 |
| | Image | 9.58 | 2.63 | 4.15 |
| | Video | 20.92 | 9.64 | 6.44 |
| | Link | 16.57 | 1.12 | 4.28 |
| Treatment | Text | 2.17 | 1.2 | 1.29 |
| | Image | 11.39 | 6.39 | 7.72 |
| | Video | 10.8 | 12.82 | 6.38 |
| | Link | 5.46 | 0.76 | 1.99 |
| *Caregiver* | Text | 2.73 | 1.96 | 2.3 |
| | Image | 15.68 | 3.95 | 5.38 |
| | Video | 19 | 8 | 1.18 |
| | Link | 5.94 | 1.5 | 2.43 |
| Impact | Text | 2.67 | 0 | 0.67 |
| | Image | 6.19 | 3.3 | 3.11 |
| | Video | 14.25 | 11.13 | 13.75 |
| | Link | 15.1 | 2.15 | 4.1 |

(*Continued*)

**Table 5.** (Continued)

| Content | Type | Likes | Shares | Comments |
|---|---|---|---|---|
| Communication Practices | Text | 2 | 1.83 | 1.67 |
| | Image | 4.94 | 3.06 | 4.44 |
| | Video | 11.4 | 7.6 | 4.4 |
| | Link | 4.67 | 0.72 | 1.4 |
| Roles and Decision Making | Text | 0 | 0 | 0 |
| | Image | 0 | 0 | 0 |
| | Video | 0 | 0 | 0 |
| | Link | 3 | 0.35 | 2.25 |
| Patient Support & Care | Text | 2.34 | 1.49 | 1.97 |
| | Image | 26.25 | 4.43 | 7.01 |
| | Video | 21.19 | 8.23 | 9.54 |
| | Link | 4.18 | 1 | 2.04 |
| Self-care | Text | 4.05 | 2.9 | 3.45 |
| | Image | 4.2 | 3.42 | 3.58 |
| | Video | 17.06 | 7.12 | 16.82 |
| | Link | 6.86 | 2.14 | 2.93 |

theories, and evidence from multiple disciplines such as human factors, marketing, engineering, sociology, and health [35]. This approach aims to actively involve different stakeholders with the intention to understand their needs and barriers towards creating meaningful, actionable, and feasible knowledge [34, 36], thereby enhancing communication and enriching the health information designed [37]. Hence, making it an ideal methodology for co-designing information in stroke caregiving.

## Study limitations

The study was focused on understanding the information-seeking behaviour, types of information available, and interaction of caregivers online through Google Insights and Content Analysis of popular social media platforms. During the analysis process, several limitations

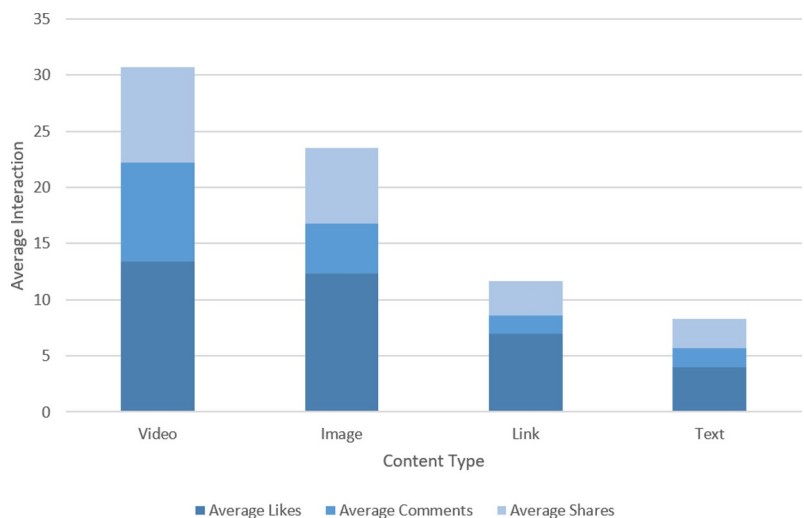

**Fig 7. Average interaction based on user likes, comments and shares for different content types.**

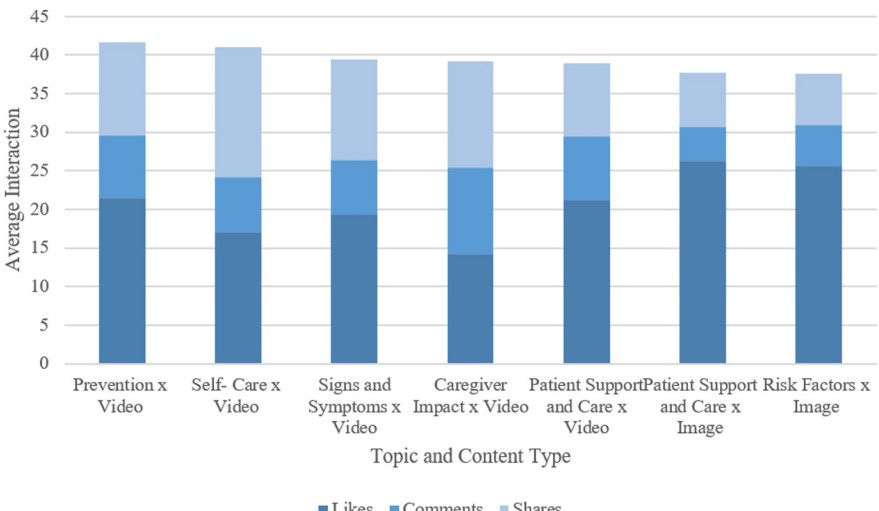

**Fig 8. Most interacted topics based on content types.**

arose. First, the inability of the scraper tool developed to extract all the posts from the social media community due to the particular restrictions by the social media platforms that monitor programs accessing social media content and blocking its access. Further, several posts were excluded during the filtration process if they did not include the target audience, i.e., the caregiver. These limitations may have resulted in several excluded posts that may have supported the caregiver during their care process. Second, the user interaction analysis considered the likes, shares, and comments of all audiences within the community as it was impossible to segregate the users based on their role. If the posts were segregated based on the type of user, the outcomes might demonstrate a difference in caregiver information needs and their level of interaction on the popular social media platforms. Third, the search criterion was limited to only English, and we are unsure if the inclusion of non-English communities may impact the outcomes of the findings. Finally, the exclusion of communities that are not publicly accessible. We excluded these communities due to ethical considerations and privacy. As a result, we are unsure if the discussions within these communities would provide a comprehensive understanding of the health information needs of caregivers and their levels of interaction.

## Conclusions

The study investigated the information-seeking behavior on Google and the content and user interaction on popular social media platforms. Findings suggest that there is a significant rise in online searches over the past ten years in stroke. The surge is indicated on both Google and social media communities. On analysis of comments designed explicitly for caregivers, topics related to the continuum of care and self-care were most engaging, especially in video-based formats. However, content creators need to understand the influences of information needs and delivery to maximize user interaction. This may be possible through co-design practices such as participatory design, which has in the past demonstrated efficient results in enhancing communication practices and enriching health information quality. Therefore, creating a deeper understanding of the caregiver and necessary information topics ensures they are prepared throughout the care process.

## Supporting information

**S1 File. Keywords searched in Google Trends and Insights.**
(XLSX)

**S2 File. Included social media communities.**
(XLSX)

**S3 File. Included community posts and user interactions.**
(XLSX)

## Author Contributions

**Conceptualization:** Elton H. Lobo, Anne Frølich, Finn Kensing, Lene J. Rasmussen, Sarah M. Hosking, Amy T. Page, Patricia M. Livingston, Sheikh Mohammed Shariful Islam, John Grundy, Mohamed Abdelrazek.

**Data curation:** Elton H. Lobo, Tara Johnson, Patricia M. Livingston, Mohamed Abdelrazek.

**Formal analysis:** Elton H. Lobo, Tara Johnson, Anne Frølich, Sheikh Mohammed Shariful Islam, Mohamed Abdelrazek.

**Investigation:** Elton H. Lobo, Tara Johnson, Amy T. Page.

**Methodology:** Elton H. Lobo, Finn Kensing, Sheikh Mohammed Shariful Islam, Mohamed Abdelrazek.

**Software:** Elton H. Lobo.

**Supervision:** Anne Frølich, Finn Kensing, Lene J. Rasmussen, Patricia M. Livingston, Sheikh Mohammed Shariful Islam, John Grundy, Mohamed Abdelrazek.

**Validation:** Elton H. Lobo, Anne Frølich, Finn Kensing, Lene J. Rasmussen, Patricia M. Livingston, Sheikh Mohammed Shariful Islam, Mohamed Abdelrazek.

**Writing – original draft:** Elton H. Lobo, Tara Johnson.

**Writing – review & editing:** Elton H. Lobo, Anne Frølich, Finn Kensing, Lene J. Rasmussen, Sarah M. Hosking, Amy T. Page, Patricia M. Livingston, Sheikh Mohammed Shariful Islam, John Grundy, Mohamed Abdelrazek.

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
