## [Decision Letter · Decision Letter 0]

16 Aug 2021

PONE-D-21-21212

Utilization of Social Media Communities for Caregiver Information Support in Stroke Recovery: An Analysis of Content and Interactions

PLOS ONE

Dear Dr. Lobo,

Thank you for submitting your manuscript to PLOS ONE. After careful consideration, we feel that it has merit but does not fully meet PLOS ONE’s publication criteria as it currently stands. Therefore, we invite you to submit a revised version of the manuscript that addresses the points raised during the review process.

We look forward to receiving your revised manuscript.

Kind regards,

Vishnu Renjith

Academic Editor

PLOS ONE

Journal Requirements:

“This study was supported through doctoral scholarships from the School of Information Technology, Deakin University, and the Department of Public Health, University of Copenhagen. Further, Prof John Grundy is supported by ARC Laureate Fellowship FL190100035, and Dr. Sarah Hosking is supported by an Alfred Deakin Postdoctoral Research Fellowship.”

Reviewers' comments:

Reviewer's Responses to Questions

**Comments to the Author**

1. Is the manuscript technically sound, and do the data support the conclusions?

Reviewer #1: Yes

Reviewer #2: No

2. Has the statistical analysis been performed appropriately and rigorously? 

Reviewer #1: Yes

Reviewer #2: Yes

3. Have the authors made all data underlying the findings in their manuscript fully available?

Reviewer #1: Yes

Reviewer #2: No

4. Is the manuscript presented in an intelligible fashion and written in standard English?

Reviewer #1: Yes

Reviewer #2: Yes

5. Review Comments to the Author

Reviewer #1: The manuscript has been nicely constructed, addressing a very relevant social aspect in stroke care. One suggestion is to add to the discussion part, is the possibility of gathering misinformation from less trusted sources in the internet and it's possible I'll effects also.

Reviewer #2: The article is interesting to read and has used a novel area of study. However, as we read the article, few queries arise in the mind and are put down below:

• Please describe the web application tool which was developed to extract information from all social media. Whether there is any process to test the reliability of the tool? How to make sure that the tool has extracted the data as the researchers intended to do and whether it was consistent across social media platforms?

• The authors have described the research approach as a mixed method approach. If you explain the research design adopted, it would be beneficial to the readers.

• Any report on geographical location of the data could be identified?

• Some of the keywords are pure medical terms and could have been searched by students of medicine, nursing or allied health professions. Is there a way to say it with certainty that the searches were made by caregivers? Common man may use terms like paralysis, weakness, not able to talk etc. for stroke.

• “The average 139 rise in the cumulative search volume was 12.4 from January 2011 to December 2020” Please explain how this value is calculated and what is the unit of the value. Explain the numbers on the ‘X’ axis of figure 1.

• Please explain how ‘Traumatic Brain Injury’ can be a keyword to identify stroke?

• In the exclusion criteria it was mentioned that sites used for Promotion of Products and Services were excluded whereas in the analysis 14 sites are included whose purpose was advertising.

• 64 (32.3%)sites were created by individuals. Whether this report could bring out the authenticity of the content on those sites?

• The content of table 5 has to be rechecked. ‘what is stroke’ query in Facebook by the reviewer generated more than one videos and plenty of likes, shares and comments. Other areas also need to be checked.

6. PLOS authors have the option to publish the peer review history of their article (what does this mean?). If published, this will include your full peer review and any attached files.

Reviewer #1: **Yes: **Shambaditya Das

Reviewer #2: No

---

## [Author Response · Author response to Decision Letter 0]

2 Sep 2021

Thank you for your feedback. We have amended the manuscript to reflect changes based on your comments.

To the editor, please replace the funding statement text with 

"This study was supported through doctoral scholarships from the School of Information Technology, Deakin University, and the Department of Public Health, University of Copenhagen. Further, Prof John Grundy is supported by ARC Laureate Fellowship FL190100035, and Dr. Sarah Hosking is supported by an Alfred Deakin Postdoctoral Research Fellowship". 

Thank you

---

## [Editor Report · Decision Letter 1]

10 Jan 2022

Utilization of Social Media Communities for Caregiver Information Support in Stroke Recovery: An Analysis of Content and Interactions

PONE-D-21-21212R1

Dear Dr. Lobo,

We’re pleased to inform you that your manuscript has been judged scientifically suitable for publication and will be formally accepted for publication once it meets all outstanding technical requirements.

Kind regards,

Vishnu Renjith

Academic Editor

PLOS ONE
---

## [Editor Report · Acceptance letter]

13 Jan 2022

PONE-D-21-21212R1 

Utilization of social media communities for caregiver information support in stroke recovery: An analysis of content and interactions 

Dear Dr. Lobo:

I'm pleased to inform you that your manuscript has been deemed suitable for publication in PLOS ONE. Congratulations! Your manuscript is now with our production department. 

Kind regards, 

on behalf of

Dr. Vishnu Renjith 

Academic Editor

PLOS ONE